# Elevation-Dependent Trends in Precipitation Observed over and around the Tibetan Plateau from 1971 to 2017

**Wenfeng Hu** [1,2]**, Junqiang Yao** [2,]*****, Qing He** [2,]***** and Jing Chen** [2]

[1] History, Culture and Tourism School, Fuyang Normal University, Fuyang 236037, China; huwf12@lzu.edu.cn
[2] Institute of Desert Meteorology, China Meteorological Administration, Urumqi 830002, China; ch_jing92@163.com
***** Correspondence: yaojq1987@126.com or yaojq@idm.cn (J.Y.); qhqinghe@163.com (Q.H.)

**Abstract:** The Tibetan Plateau (TP) are regions that are most sensitive to climate change, especially extreme precipitation changes with elevation, may increase the risk of natural disasters and have attracted attention for the study of extreme events in order to identify adaptive actions. Based on daily observed data from 113 meteorological stations in the Tibetan Plateau and the surrounding regions in China during 1971–2017, we calculated the annual total precipitation and extreme precipitation indices using the R ClimDex software package and explored elevation-dependent precipitation trends. The results demonstrate that the annual total precipitation increased at a rate of 6.7 mm/decade, and the contribution of extreme precipitation to total precipitation increased over time, and the climate extremes were enhanced. The annual total, seasonal precipitation, and precipitation extreme trends were observed in terms of elevation dependence in the Tibetan Plateau (TP) and the surrounding area of the Tibetan Plateau (TPS) during 1971–2017. There is growing evidence that the elevation-dependent wetting (EDWE) is complex over the TP. The trends in total precipitation have a strong dependence on elevation, and the EDWE is highlighted by the extreme precipitation indices, for example, the number of heavy precipitation days (R10) and consecutive wet days (CWD). The dependence of extreme precipitation on elevation is heterogeneous, as other extreme indices do not indicate EDWE. These findings highlight the precipitation complexity in the TP. The findings of this study will be helpful for improving our understanding of variabilities in precipitation and extreme precipitation in response to climate change and will provide support for water resource management and disaster prevention in plateaus and mountain ranges.

**Keywords:** elevation-dependent wetting (EDWE); extreme precipitation; precipitation; tibetan plateau





## 1. Introduction

IPCC (2007) reported that the air temperature of the global surface has increased by 0.75 °C during the last century [1]. Under the context of global warming, mountainous regions may be more sensitive to warming than other land areas at the same latitude [2,3]. Studies have reported that the warming rate in mountainous areas was higher than that in low-altitude areas [2,4–8]. Most climate models revealed that mountainous areas exhibited rapid warming compared to other land types, and the simulation results were more consistent than the observed results [9]. Many previous studies have quantified and identified elevation-dependent warming, and strong evidence has been gathered that suggests that warming occurs more rapidly at higher elevations than at lower elevations [7,10]. This phenomenon may exhibit regional differences, and researchers have evaluated climate change in the mountains of the American West [11], the European Alps [12], the Andes Mountains [13], and the mountains of eastern Australia [14] to determine region-specific effects. It was found that there were different degrees of warming in mountainous areas, but other researchers have reported conflicting findings regarding elevation-dependent warming. Oyler et al. [11] pointed out that higher altitude areas are warming more rapidly than lower altitude areas in mountain ranges in the western United States. However,

Tudoroiu et al. [12] indicated that warming occurred at both high and low elevations in the Eastern Italian Alps, but it was less obvious at high elevations. Russell et al. [13] denoted that there was no evidence for either elevation-dependent warming or regional amplification of warming, on average, in the tropical Andes of South America. Although there have been different perceptions regarding elevation-dependent warming, warming in mountainous areas is still recognized by scholars. Furthermore, other effects caused by warming in mountainous areas have become the focus of research. Warming causes increased evapotranspiration and intensifies the hydrological cycle [15,16], which further impacts water resources and ecosystem services [11]. Specifically, precipitation in mountainous areas has attracted the attention of researchers. Studies have focused on the relationship between precipitation and altitude as well as current and future impacts [17–21], especially in mountainous areas. Generally, the results have indicated that precipitation increases continuously with elevation up to the mountain peaks or an elevation below the top, above which precipitation amounts did not increase any more considering the amplification of elevation-dependent wetting at high elevations, studies have focused on the precipitation trends along elevation gradients in the mountains [22–25]. Studies on elevation dependence of precipitation changes are important for impact assessments including those on the glaciological dynamics and hydrological cycles in the high-elevation regions, but research on the relationship between precipitation and elevation was insufficient in mountainous areas.

The Tibetan Plateau (TP), which is considered the "Third Pole," is an area of water resource conservation and major runoff production. It is an important ecological barrier and the birthplace of modern glacier research in China. With an increase in population and economic development, the ecological environment has deteriorated. The TP has experienced significant climate change impacts, including changes in the cryosphere and elevation-dependent wetting [26–31]. The TP is currently in the period of warming and wetting [27,28,32–34] and exhibits elevation-dependent wetting [7,10]. Elisa Palazzi et al. [35] reported a nonlinear relationship between rising temperatures and elevation, while You et al. [36]) pointed out that enhanced sensitivity to extreme temperatures at higher elevations was not apparent considering the background warming in this area. However, the decreased rate of near-surface wind speed was enhanced at high elevations, and the near-surface wind speed exhibits more substantial changes in high-elevation areas than in low-elevation areas [37]. Some studies have reported on elevation-dependent effects of climate change on vegetation coverage in the TP [38]). Although changes in summer precipitation showed an upward trend with increasing elevation [39]), the elevation-dependent change exhibited significant regional differences in the TP and the surrounding area of the Tibetan Plateau (TPS). No significant change was observed over the southeastern TPS. A decreasing tendency was observed with the increased elevation appears in the interior of TPS with an elevation higher than 4000 m, and an increasing tendency was observed elsewhere [40].

Studies on the elevation-dependent effects of climate change are important for various environmental assessments, including those on hydrological cycles or glaciological dynamics in the high-elevation areas of the TPS. It is noted that the study on elevation-dependent precipitation is of great significance to the advance and retreat of modern glaciers, water resources management and disaster prevention and cure in the TP. However, research on the elevation-dependent wetting remains insufficient—studies on elevation-dependent precipitation trends typically analyze only in the summer or the wet season [39,40]. Thus far, a systematic study has not been conducted on elevation-dependent precipitation, and it is not clear whether elevation-dependent trends are observed for extreme precipitation. The present study focused on the following: (1) spatiotemporal changes in precipitation, especially changes in extreme precipitation; (2) the relationship between precipitation changes and elevation; and (3) the relationship between extreme precipitation trends and elevation. Details of data and methodology are provided in Section 2. Section 3 describes the spatiotemporal changes in precipitation and the relationship between precipitation

(extreme precipitation) trends and elevation. The findings are discussed in Section 4, and our conclusions are presented in Section 5.

## 2. Data and Methodology

The TP and parts of the neighboring Yunnan, Sichuan, and Gansu provinces were selected as the study region (Figure 1). In total, 113 meteorological stations were selected, which are mainly distributed in the southeast of the TP and the TPS Figure 1a. The altitudes of these sites range between 1000–5000 m; the detailed characteristics of the selected sites are shown in Figure 1b. The daily meteorological data from 113 weather stations were provided by the China Meteorological Administration (CMA) (http://data.cma.cn) assessed on 15 February 2020. After accounting for the missing data and comparing the length of the recording period, we selected the period of 1971–2017. Data quality control and calculation of the extreme precipitation index from the original datasets were undertaken using the R ClimDex software http://etccdi.pacificclimate.org/software.shtml (assessed on 6 October 2020). The detailed calculation process is shown in ClimDex (1.0). User Manual [41], (Full details are given in Appendix A). Data quality control and the calculation process were conducted in strict accordance with the study requirements.

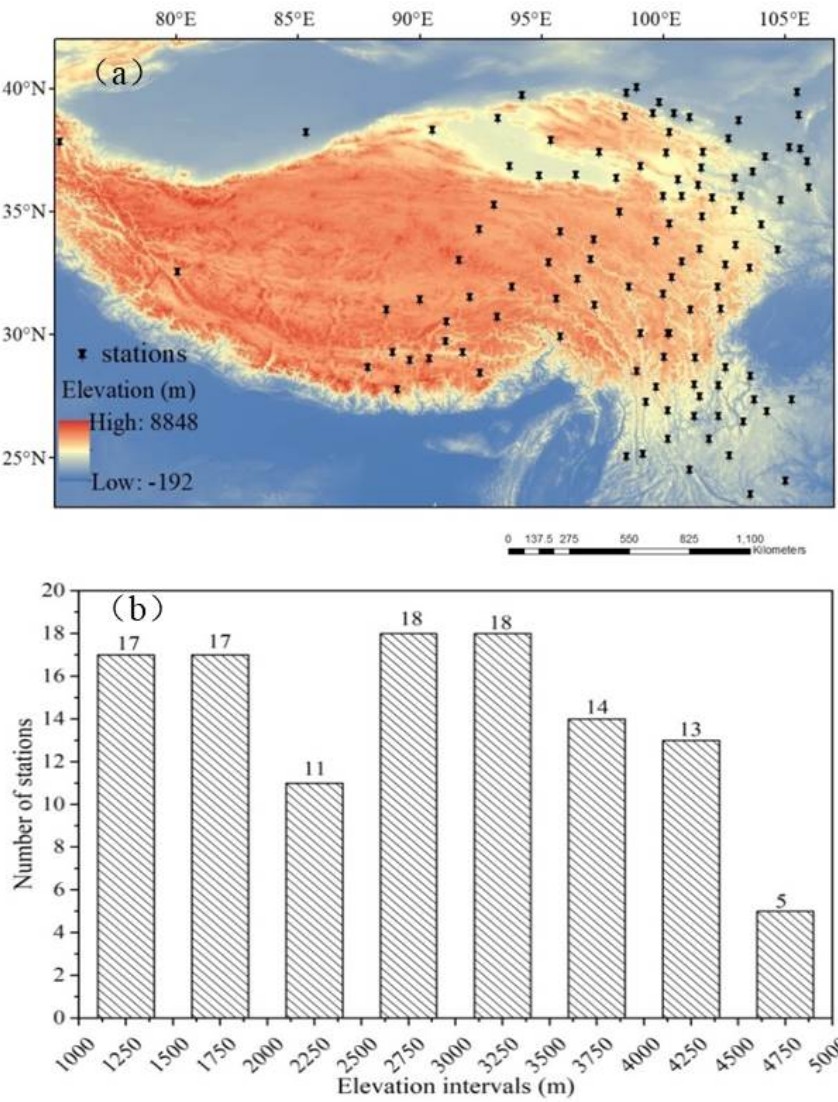

**Figure 1.** Topographic map of the TP and TPS with 113 weather stations from CMA (**a**) And the elevation distribution of stations with number of stations for each elevation interval is shown (**b**).

In this study, we used extreme precipitation indices to detect and analyze the changes in extreme precipitation. There are many definitions of extreme precipitation, and Nichols et al. [42] presented a series of extreme precipitation indices. We used the methods presented by the World Climate Research Programme (WCRP) project operated by the US Climate Variability and Predictability (CLIVAR) expert team on climate change detection [43]. The indices describe different aspects of the precipitation process; these were defined and used to analyze and detect precipitation extremes and variations in our study (Table 1). All selected indices can be roughly divided into four categories [44,45].

**Table 1.** Definitions of the precipitation indices.

| ID | Indicator Name | Definitions | Units |
|---|---|---|---|
| | | absolute indices | |
| RX1day | Max 1-day precipitation amount | Monthly maximum 1-day precipitation | mm |
| SDII | Simple daily intensity index | Annual total precipitation divided by the number of wet days (defined as PRCP >=1.0 mm) in the year | mm/day |
| PRCPTOT | Annual total wet-day precipitation | Annual total PRCP in wet days (RR >= 1 mm) | mm |
| | | threshold indices | |
| R10 | Number of heavy precipitation days | Annual count of days when PRCP >= 10 mm | day |
| | | duration indices | |
| CDD | Consecutive dry days | Maximum number of consecutive days with RR < 1 mm | day |
| CWD | Consecutive wet days | Maximum number of consecutive days with RR >= 1 mm | day |
| | | percentile-based threshold indices | |
| R95p | Very wet days | Annual total PRCP when RR > 95th percentile | mm |

The linear regression method shown below was adopted to analyze changes in extreme precipitation (or extreme precipitation indices)

$$y = ax + b \qquad (1)$$

where y represents the extreme precipitation (or extreme precipitation indices), x denotes the time (in years), a is the slope (changing rates of extreme precipitation), and b is the intercept. The regression parameters a and b were determined using the least-squares fitting method. The statistical significance of the changing rates was evaluated using the t-test. And the statistics of other extreme precipitation indices are shown in Table 2.

**Table 2.** Number of stations with significance of trends for extreme precipitation indices (*p*-value).

| ID | $p < 0.01$ | $0.01 < p < 0.05$ | $0.05 < p < 0.1$ | $0.1 < p$ |
|:---:|:---:|:---:|:---:|:---:|
| CDD | 4 | 24 | 22 | 63 |
| CWD | 8 | 18 | 21 | 66 |
| PRCTOT | 6 | 14 | 26 | 67 |
| SDII | 5 | 17 | 25 | 65 |
| R95p | 10 | 25 | 27 | 51 |
| R10 | 5 | 18 | 30 | 60 |
| RX1day | 6 | 23 | 35 | 39 |

## 3. Results

### 3.1. Spatiotemporal Changes in Precipitation

The annual total PRCPTOT (RR $\geq$ 1 mm) was found to increase from 1971 to 2017 Figure 2c, and the rate of increase was 6.7 mm/decade (slope: 0.67; R: 0.3). The spatial changes in PRCPTOT are shown in Figure 2a. Spatial heterogeneity in the PRCPTOT was observed over the study area. The PRCPTOT gradually decreased from southeast to northwest. The spatial distributions of long-term (1970–2017) PRCPTOT trends are shown in Figure 2b. Notable regional differences were observed; most stations exhibited sharply increasing trends, while others in the southeast and at the edge of the plateau exhibited decreasing trends. Interestingly, the PRCPTOT at some stations in high-elevation areas increased more rapidly than those in low-elevation areas, and the PRCPTOT at some stations in low-elevation areas decreased.

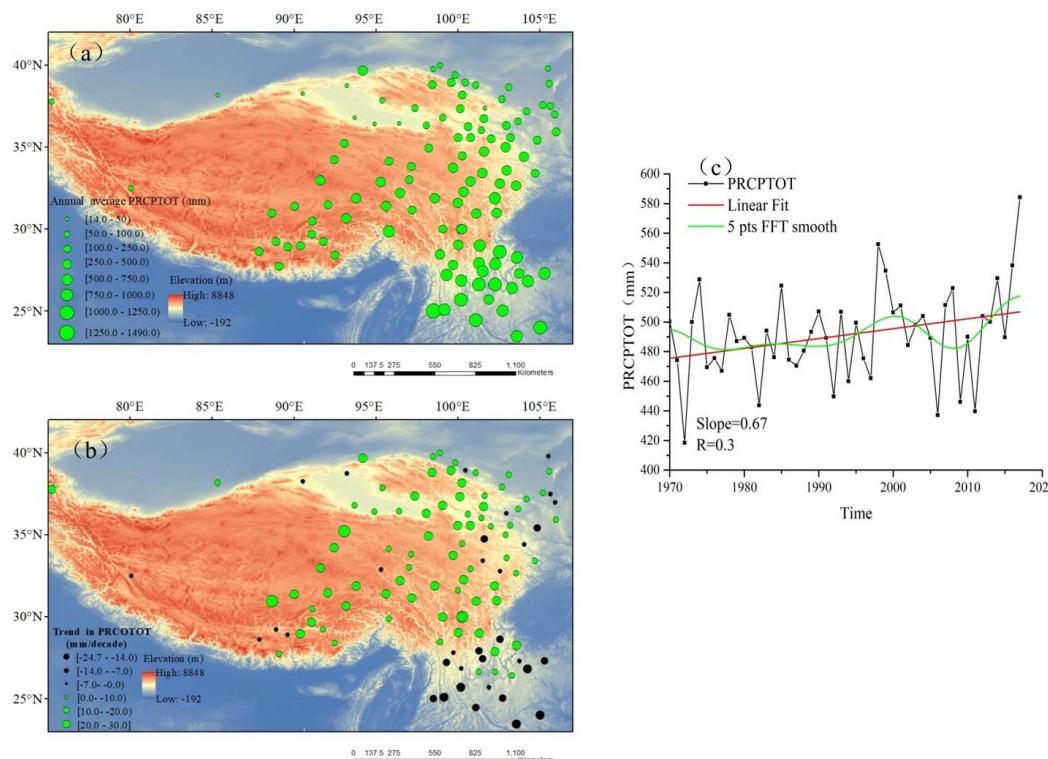

**Figure 2.** Spatial distribution of the averaged and long-term climatological trends in annual mean PRCPTOT (**a**,**b**) and changes in annual mean PRCPTOT (**c**) in the TP and TPS during 1971–2017.

In the study area, annual precipitation was 505 mm. The percentage of average monthly precipitation to annual precipitation is shown in Figure 3. Monthly precipitation was unevenly distributed throughout the year. July and December experienced the highest and the lowest precipitation, accounting for 21.2% and 0.7% of the annual precipitation, respectively. Seasonal precipitation was the greatest in summer (June, July, and August)

and the least in winter (December, January, and February), accounting for 57.4% and 2.8% of the annual precipitation, respectively. Precipitation in spring (March, April, and May) and autumn (September, October, and November) accounted for 17.2% and 22.6% of the annual precipitation, respectively.

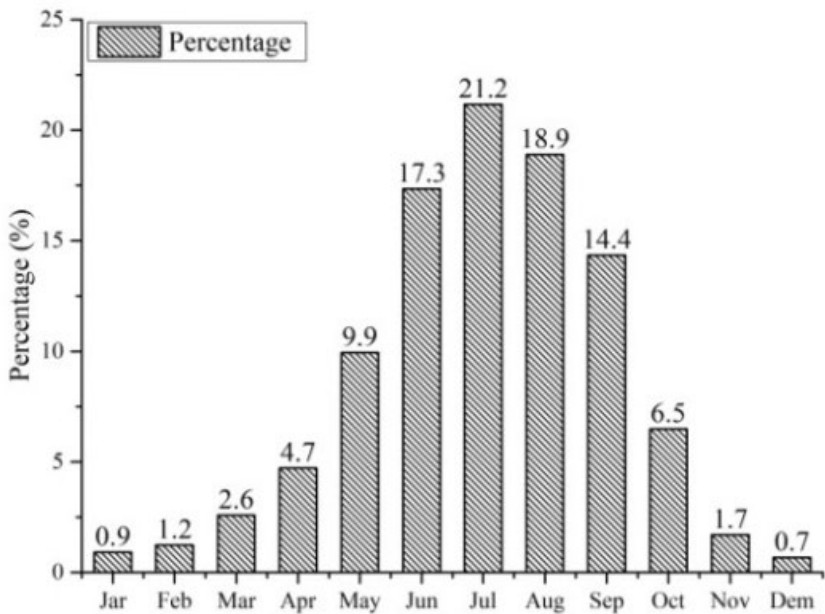

**Figure 3.** Yearly cycles in annual precipitation percentage (%) in the TP and TPS.

### 3.2. Changes in Extreme Precipitation

The total extreme precipitation (R95p) and percentage of extreme precipitation in total precipitation changed with time, as shown in Figure 4; both increased during the study period, but different increase rates were observed. Hence, in the context of global warming, the total amount of extreme precipitation and the contribution of extreme precipitation to total precipitation increased along with precipitation in the study area. The regional climate extreme had enhanced over time.

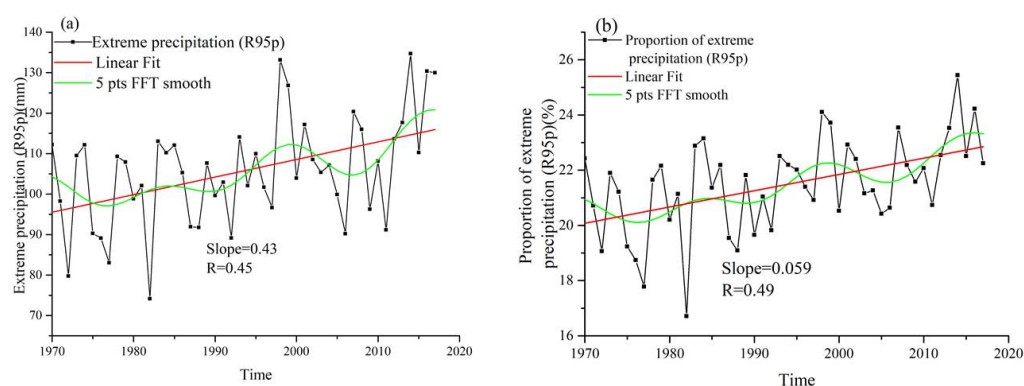

**Figure 4.** Changes in total extreme precipitation (R95p) (**a**) and extreme precipitation percentage change (**b**) in the TP and TPS during 1971–2017. The rate of increase was 4.3 mm/decade slope (0.43; R: 0.45; Figure 4a) for extreme precipitation, and the rate of increase was 0.59%/decade (slope: 0.0059; R: 0.49; Figure 4b) for extreme precipitation percentage.

The spatial trends of R95p are shown in Figure 5a. Spatial heterogeneity in R95p was observed in the study area. R95p gradually decreased from southeast to northwest, which was consistent with the spatial variation trend of precipitation. The spatial distribution of the R95p was consistent with the spatial distribution of the annual total PRCPTOT (as Figures 2a and 5a). Notable regional differences were observed; most stations exhibited

significant increasing trends, while 33 stations in the study area exhibited decreasing trends. These stations were relatively concentrated in the southeast of the TP. This also means that under the background of global warming, it was more complicated in the trend of extreme precipitation in the plateaus and mountainous areas. Maybe this indicates that warming-related changes vary according to region.

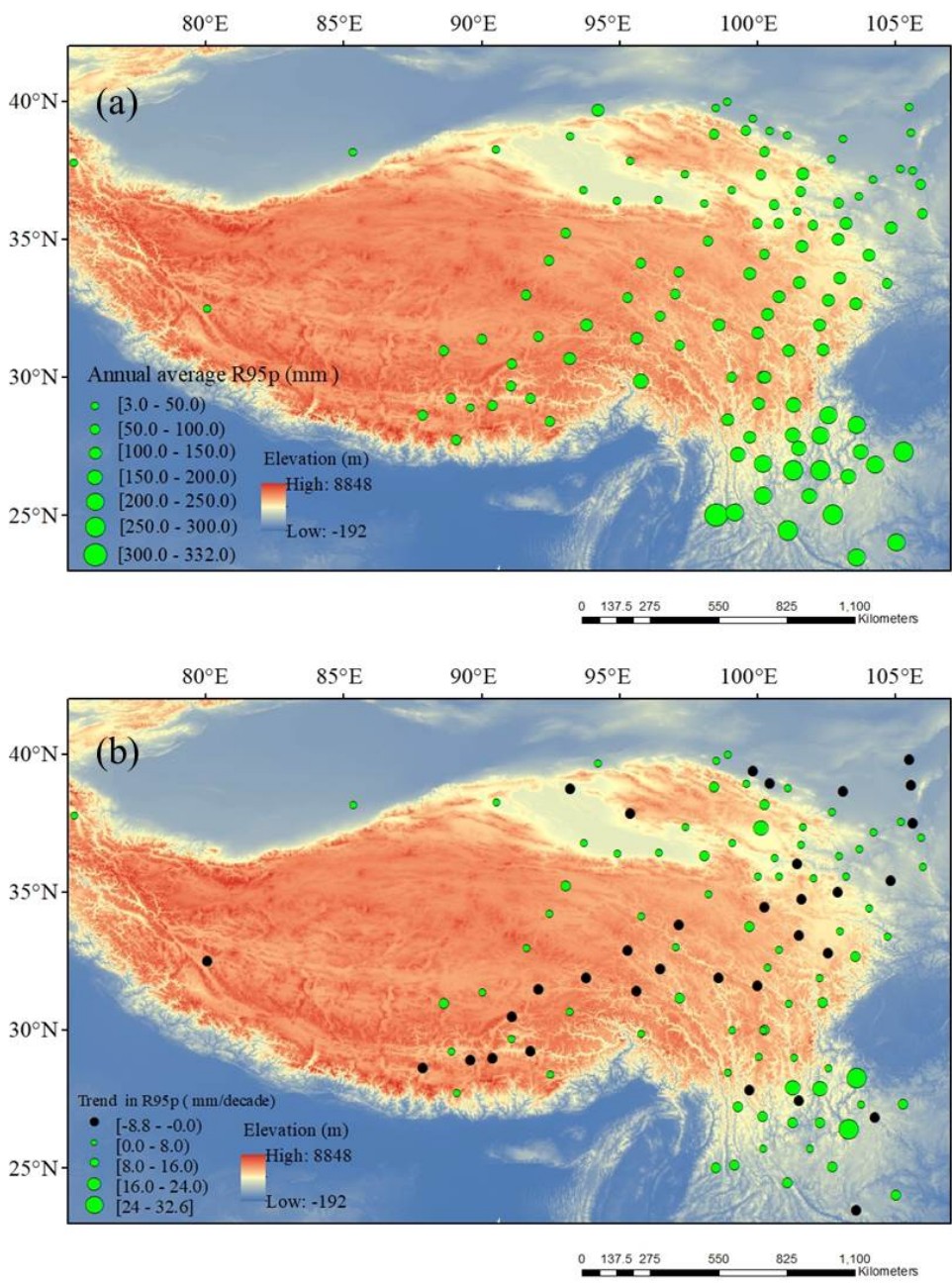

**Figure 5.** Spatial distributions of the averaged extreme precipitation (R95p) (**a**) and long-term climatological trends in annual average extreme precipitation (R95p) (**b**) in the TP and TPS during 1971–2017. (35 stations passed the 95% significance test in (**b**)).

### 3.3. Relationship between Precipitation Trends and Elevation

The relationship between the annual total PRCPTOT trend and elevation was analyzed using linear regression (Figure 6a). The PRCPTOT increased by 0.49 mm/decade/100 m (slope: 0.0049; R: 0.48; $p < 0.01$) for elevations ranging from 1000–5000 m, showing an upward trend with increasing elevation during 1970–2017. The scatter plot (Figure 6a) of the relative changes with elevation illustrates elevation dependence by a correlation

coefficient of 0.48, which is statistically significant at the 95% confidence level. Furthermore, 10 pts FFT smooth processing was applied to the scatter plot of the relationship between the PRCPTOT trend and elevation, and it was found that the dependence of PRCPTOT on elevation is not a simple linear relationship. To further analyze the sensitivity of elevation dependence, changes in annual total PRCPTOT trends with increasing elevation at intervals of 500 m were explored for the study period (Figure 6b). The annual total PRCPTOT showed a distinct increasing trend with the elevation gradient, but the trend fluctuated with differences in elevation (Figure 6b).

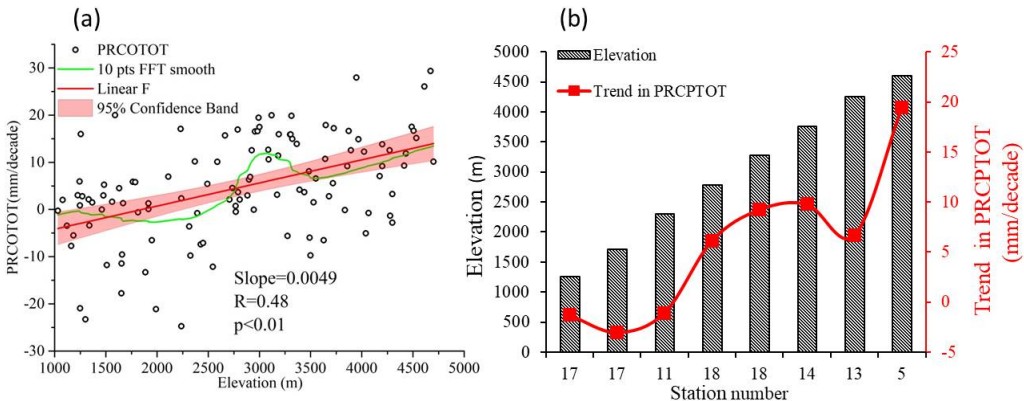

**Figure 6.** (**a**) Scatter plots of trends (mm/decade) by elevation for PRCPTOT and (**b**) elevation dependence of PRCPTOT trends (mm/decade) in the TP and TPS during 1971–2017. (The numbers in b horizontal axis represent the stations in each 500 m elevation band, and trend magnitude is plotted on the y axis, the trends are the means over individual elevation bins).

Due to the obvious seasonal distribution of precipitation in the study area (Figure 2), the relationship between the seasonal PRCPTOT trends and elevation and the sensitivity of elevation dependence were analyzed. The scatter plot showing the relationship between seasonal total PRCPTOT trends and elevation is shown in Figure 7. Seasonal PRCPTOT trends exhibited different characteristics according to varying elevation during 1970–2017. PRCPTOT trends in summer and autumn exhibited an upward trend with elevation, and PRCPTOT increased by 0.23 mm/decade/100 m (slope: 0.0023; R: 0.30; $p < 0.01$) and 0.067 mm/decade/100 m (slope: 0.00067; R: 0.15; $p > 0.05$), respectively, as indicated by correlation coefficients of 0.30 and 0.15, respectively, which is statistically significant at the 95% confidence level only in summer. However, PRCPTOT trends in winter also increased with elevation at a rate of 0.005 mm/decade/100 m (slope: 0.00005; R: 0.05; $p > 0.05$), as indicated by a correlation coefficient of 0.05, but it did not pass the significance test. Interestingly, PRCPTOT trends in spring exhibited an obvious downward trend with elevation, decreasing by $-0.06$ mm/decade/100 m (slope: $-0.0006$; R: $-0.34$; $p < 0.01$), as indicated by a correlation coefficient of $-0.34$, and which is statistically significant at the 95% confidence level. Also, 10 pts FFT smooth processing was applied to the scatter plot of the relationship between the seasonal PRCPTOT trend and elevation; differences in trends were observed at different elevations.

The trend changes in the seasonal PRCPTOT with the elevation increase at 500-m wide elevation bands are shown in Figure 8. PRCPTOT trends in summer and autumn exhibited an increasing tendency along the elevation gradient (Figure 8), although the strengths of elevation dependence differed at various elevations. This dependence was prominent in the elevation range of 1000–3500 m; however, a decline in the trend was observed in the range of 3500–5000 m. However, the change in spring PRCPTOT exhibited a downward trend with elevation and had no obvious dependence on elevation in winter. Hence, the elevation dependence of the precipitation trend varied in different seasons and at different elevations. Thus, the seasonal PRCPTOT presented a different fluctuation in trend along the elevation and a simple increasing or decreasing trend with elevations was not apparent.

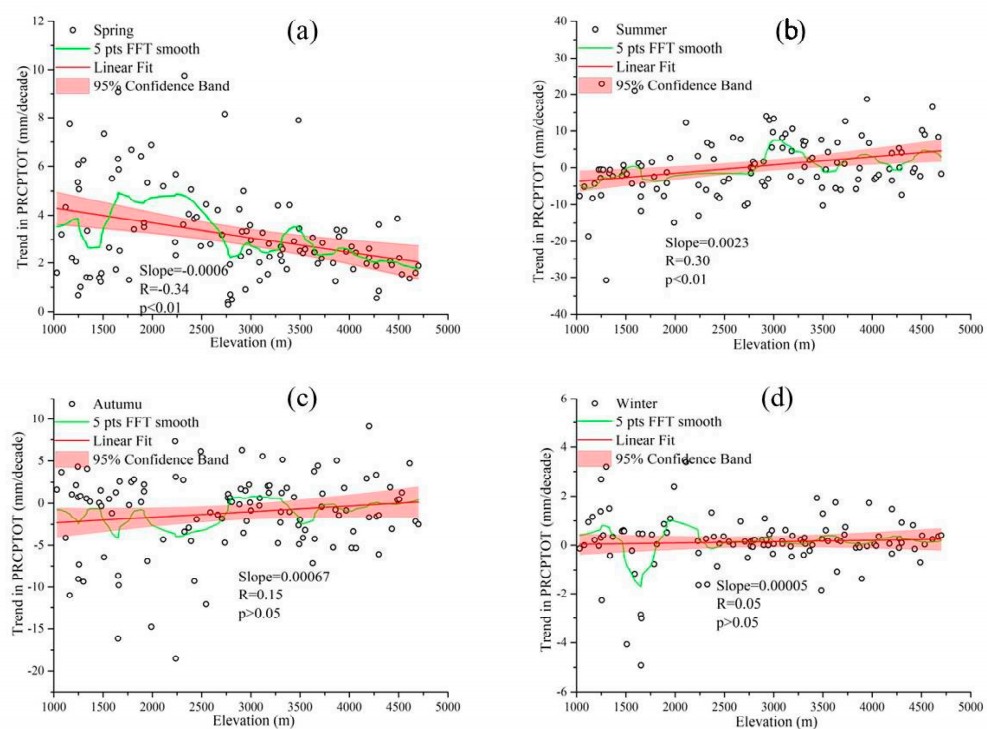

**Figure 7.** Scatter plots of trends (mm/decade) by elevation for seasonal total precipitation ((**a**) spring; (**b**) summer; (**c**) autumn; (**d**) winter) in the TP and TPS during 1971–2017.

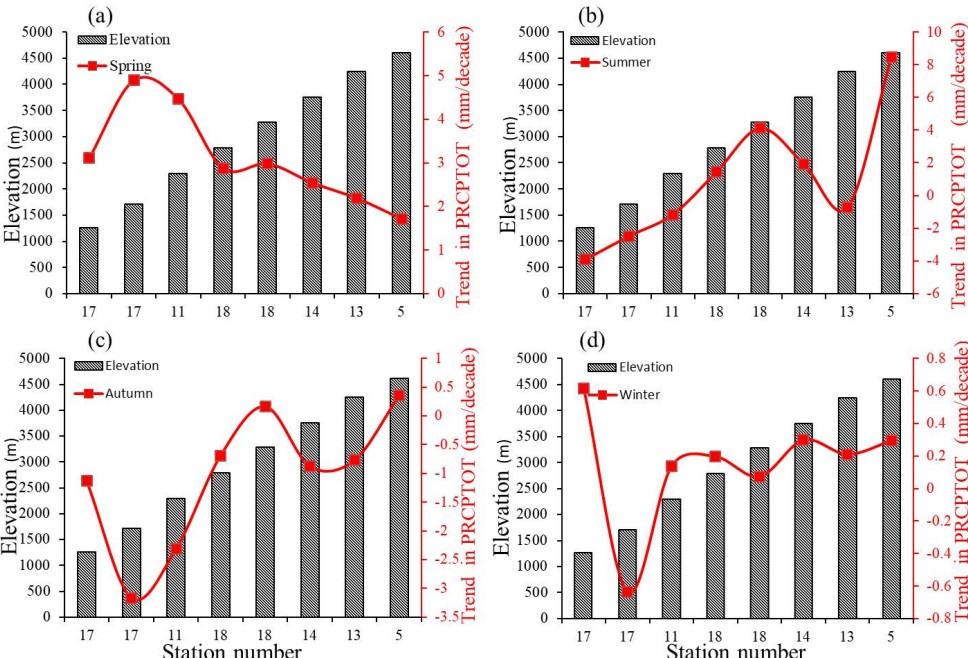

**Figure 8.** Elevation dependence of seasonal total precipitation trends (mm/decade) ((**a**) spring; (**b**) summer; (**c**) autumn; (**d**) winter) in the TP and TPS during 1971–2017. (The numbers in horizontal axis represent the stations in each 500 m elevation band, and trend magnitude is plotted on the y axis, the trends are the means over individual elevation bins).

### 3.4. Relationship between Extreme Precipitation Trend and Elevation

Figure 9 shows the relationship between the change trends in extreme precipitation indices and elevation during 1970–2017, and the correlation coefficients between them are statistically significant at the 95% confidence level. There were negative relevant relations

between elevation and consecutive dry days (CDD), R95p, Simple daily intensity index (SDII), and Max 1-day precipitation amount (RX1day) trends, and positive correlations were observed for consecutive wet days (CWD) and R10 (Figure 9). The CDD decreased by $-0.029$ day/decade/100 m (slope: $-0.00029$; R: $-0.078$; $p > 0.05$), but it did not pass the significance test, and the CWD increased by 0.007 day/decade/100 m (slope: 0.00007; R: 0.24; $p < 0.01$) for elevation ranging from 1000–5000 m, which is statistically significant at the 95% confidence level. Although the trends were negligible, it was also indicated that as the elevation increased, the sensitivity to drought weakened while that to wetting increased. R95p and SDII trends decreased with elevation at the rates of $-0.114$ mm/decade/100 m (slope: $-0.00114$; R: $-0.19$; $p > 0.05$) and $-0.0035$ mm/day/decade/100 m (slope: $-0.000035$; R: $-0.28$; $p < 0.01$), respectively. Which is statistically significant at the 95% confidence level only in SDII. This indicates that increases with elevation, sensitivity to total extreme precipitation, and intensity of extreme precipitation events are declining. The R10 increased by 0.016 day/decade/100 m (slope: 0.00016; R: 0.35; $p < 0.01$) and the RX1day decreased by $-0.021$ mm/day/decade/100 m (slope: $-0.00021$; R: $-0.17$; $p > 0.05$), Which is statistically significant at the 95% confidence level only in R10. Hence, extreme precipitation frequency increased with the elevation over time, and the sensitivity to the annual maximum 1-day precipitation was found to be declining.

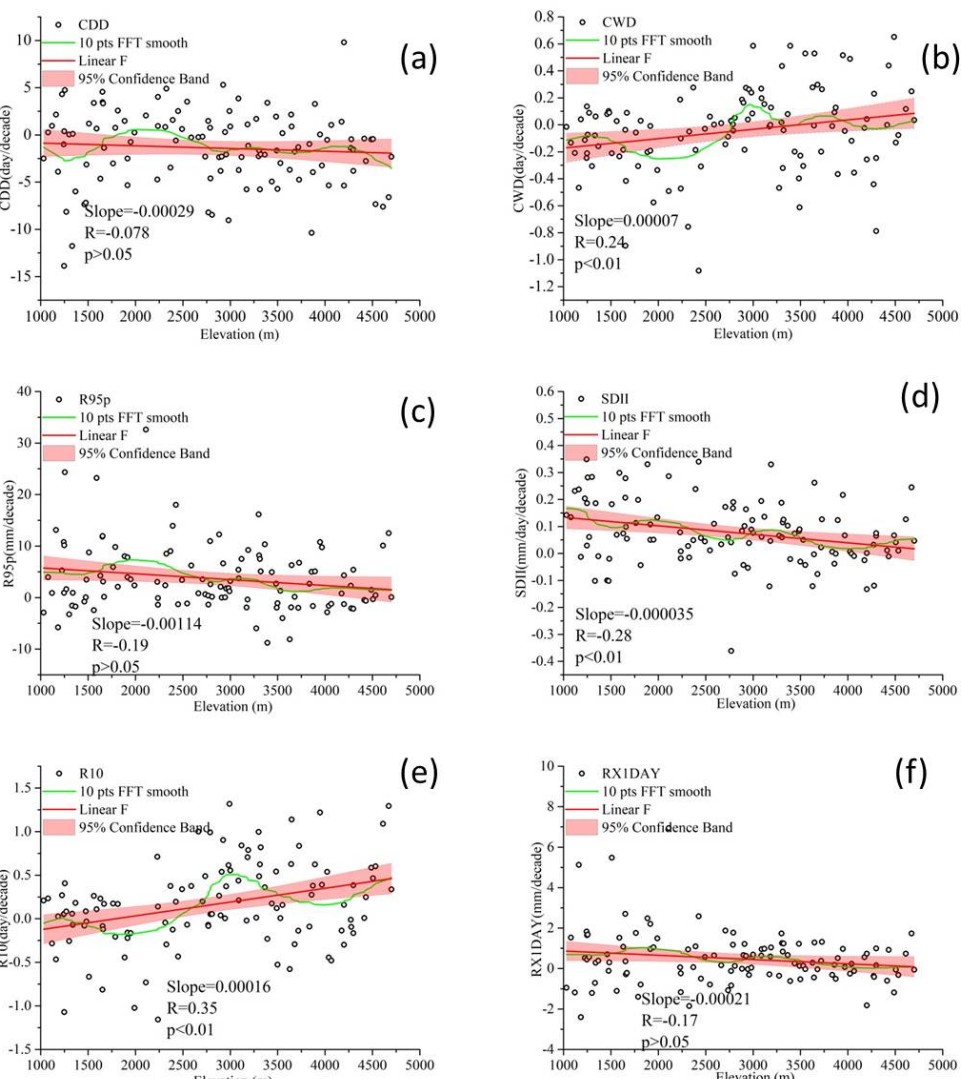

**Figure 9.** Scatter plots of trends (mm/decade) by elevation for extreme precipitation ((**a**) CDD; (**b**) CWD; (**c**) R95p; (**d**) SDII; (**e**) R10; (**f**) RX1day) in the TP and TPS during 1971–2017.

Changes in the average magnitude of each extreme precipitation index with increasing elevation at 500-m intervals and the number of weather stations in each gradient are shown in Figure 10. Each extreme precipitation index shows fluctuations at different elevations, and a simple increase or decrease with elevation was not apparent. For CDD trends, a positive correlation with elevation was observed at 1000–2000 m and a negative correlation at 2000–5000 m. However, CWD trends were more complicated; negative correlations with elevation were observed at 1000–2000 m, 2500–3000 m, and 3500–4000 m, and positive correlations were observed at 2000–2500 m, 3000–3500 m, and 4000−4500 m. Positive correlations with elevation for R95p trends were observed at 1000–2000 m, 2500–3000 m, and 4000–4500 m, and negative correlations were observed at 2000–2500 m, and 3000–4000 m. For SDII trends, negative correlations with elevation were observed at 1000–2500 m and 3000–4000 m, and positive correlations were observed at 2500–3000 m and 4000–4500 m. For R10, negative correlations with elevation were observed at 1000–1500 m and 3000–4000 m, and positive correlations were observed at 1500–3000 m and 4000–4500 m. For RX1day, positive correlations with elevation were observed at 1000–1500 m, 2500–3000 m, and 4000–4500 m and negative correlations were observed at 1500–2500 m and 3000–4000 m. These findings indicate that elevation dependence did not exhibit a linear relationship.

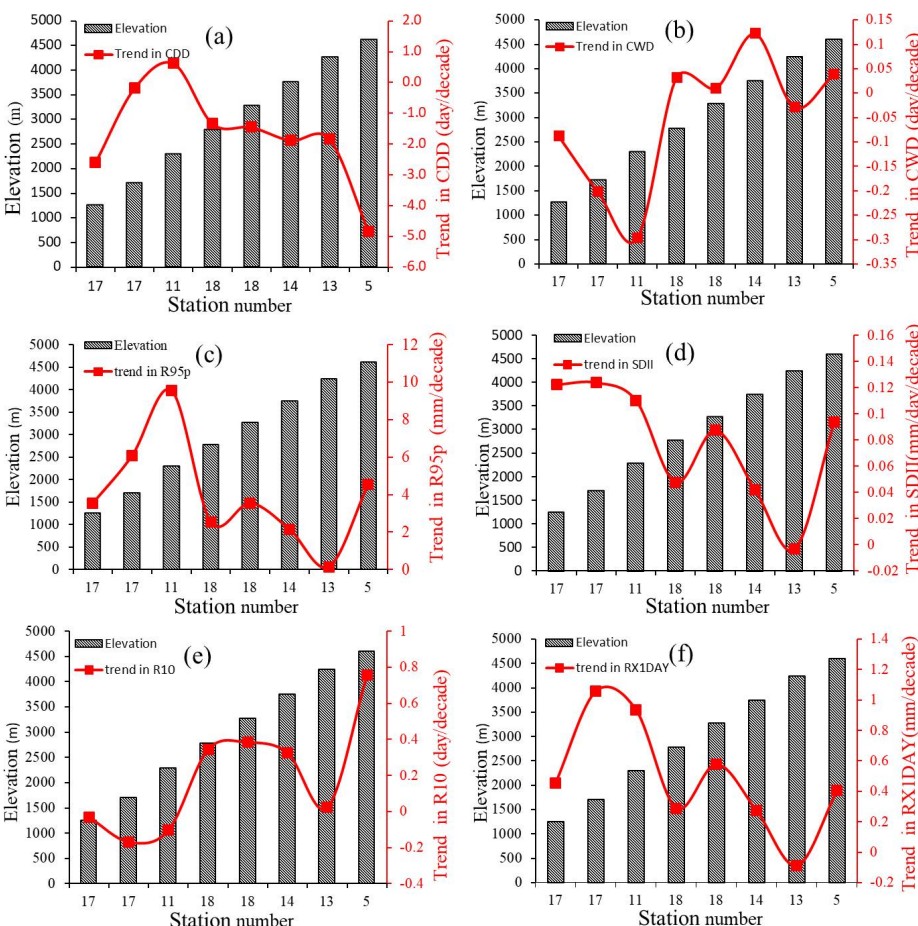

**Figure 10.** Elevation dependence of extreme precipitation trends (mm/decade) ((**a**) CDD; (**b**) CWD; (**c**) R95p; (**d**) SDII; (**e**) R10; (**f**) RX1day) in the TP and TPS during 1971–2017, (The numbers in horizontal axis represent the stations in each 500 m elevation band, and trend magnitude is plotted on the y axis, the trends are the means over individual elevation bins).

## 4. Discussion

Precipitation exhibits complex spatial behavior, which is closely related to the topography of the underlying surface [46,47] and the topographic enhancement of precipitation [48]). Kim et al. [49] studied precipitation in the Sierra Nevada and demon-

strated that the simulated seasonal total precipitation generally increased with elevation. Rowe et al. [22] found that precipitation events at lower elevations had greater intensity (rain rate) but were less frequent than those over the Sierra Madre Occidental. Beusekom et al. [25] studied the rainfall in Northeastern Puerto Rico and indicated that the precipitation in the driest months of each wet season increased rapidly in high-altitude areas; similar patterns in monthly precipitation were observed throughout the year but only at half rate. In future climate simulations, it is expected that precipitation at high (above 4000 m) and low (1500 m or below) elevations will increase in the TP, and the increase in precipitation at high elevations will be caused by the increase in evaporation from the surface [24]. However, it has also been reported that the precipitation decreases with elevation only in the hinterland of the TP with elevations above 4000 m; little to no change is observed over the southeastern area of the TP and TPS, while the increase in precipitation is noted in other regions [42].

Globally averaged surface air temperature has increased by 0.81 °C from 1961 to 2018. However, changes in the spatial patterns were heterogeneous, with the rate of land warming being more than double that of the ocean [6]. Several studies have proposed that some mountainous regions experienced warming more rapidly than other land types [4–6,10,50–52]. In our study, the annual average temperature in the region increased from 1971 to 2017 (Figure 11c), and the rate of increase was 0.33 °C/decade (slope: 0.033; R: 0.77). Spatial changes in the annual average temperature are shown in Figure 11a. Spatial heterogeneity in the annual average temperature was observed in the study area. The annual average temperature gradually decreased from the surroundings toward the interior of the plateau, and the annual average temperature was lower at high altitudes than at low altitudes. The spatial distributions of long-term (1970–2017) annual average temperature trends are shown in Figure 11b. A significant upward trend was observed for all but three stations, which exhibited a downward trend. In particular, it showed an opposite pattern in the spatial distribution of temperature and its upward trend. A slower rising trend in the high-temperature area and a faster rising trend in the low-temperature area were observed, and the upward trend on the plateau was stronger than in the TPS. Pepin et al. [10] reviewed elevation-dependent warming (EDW) from observation and model results in mountainous areas, especially in the TP. In our study, it is also found that there is an obvious elevation-dependent warming (EDW) in this area (as shown in Figure 11d,e). On the whole, the annual average temperature of the TP and TPS shows an upward trend from 1971 to 2017, and also has obvious elevation-dependent warming (EDW).

With the increase in temperature, the snow cover, ice sheets, and permafrost thaws accelerate melting in higher elevation areas and increase local atmospheric water vapor, local air humidity increases, and annual precipitation is changing at a higher rate than at lower altitudes According to the Clausius–Clapeyron relationship, intensified warming will cause an increase in the water-holding capacity of the atmosphere and lead to increased precipitation [53–56]. Donat et al. [57,58] also suggested that the warming rate is significantly statistically correlated with precipitation intensification. In mountainous environments, the EDW can strengthen the change rate of the local hydrological cycle [10,59] and increase atmosphere moisture and precipitation. Yao et al. [60] firstly proposed the concept of elevation-dependent wetting (EDWE), indicating that the precipitation-increasing trend is amplified with elevation. Evidence for EDWE is found in the observed precipitation trends and elevation relationships in arid regions of China [60]. Li et al. [39] reported elevation dependence of trends in summer precipitation over the TP.

In our study, the annual and seasonal precipitation trends and the precipitation extreme trends were observed in terms of elevation dependence in the TP and TPS during 1971–2017. There is growing evidence that the EDWE is widespread on the annual total precipitation over the TP and TPS, but there were different relationships between seasonal precipitation trends and elevation at different altitudes. Meanwhile, the dependence of extreme precipitation on elevation is heterogeneous, the EDWE is highlighted by the

extreme precipitation indices in R10 and CWD, as other extreme indices do not indicate EDWE. These findings highlight the precipitation complexity in the TP and TPS. This may be due to the complex precipitation mechanism in the Tibetan Plateau and its surrounding areas. Summer precipitation accounts for more than 50% of annual total precipitation at most stations. Horizontally, precipitation decreases from the southeast to the northwest as a result of the prevailing atmospheric circulations combined with the effects of local elevation and terrain. The large-scale weather systems determine the regional patterns of precipitation over the TP and TPS, but the complex topography ought to exert significant influence through localizing and distributing precipitation, thus affecting the local spatial patterns of precipitation, the influence factors of extreme precipitation are more complex.

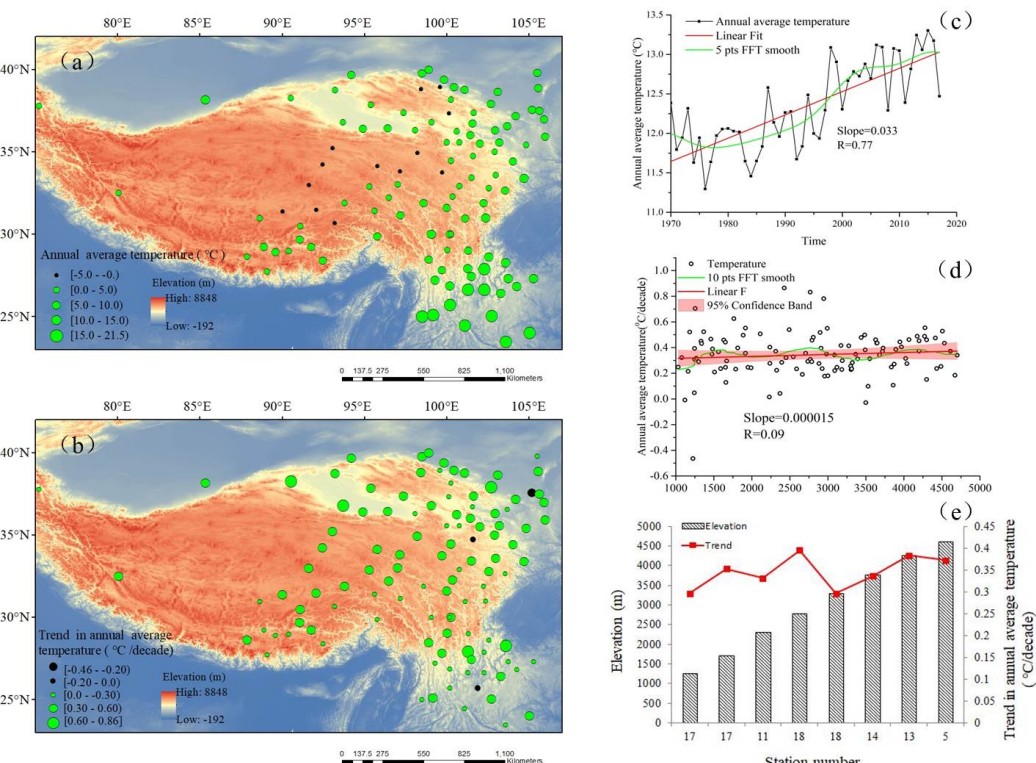

**Figure 11.** Spatial distribution of the averaged and long-term climatological trends in annual mean temperature (**a**) and (**b**), changes in annual mean temperature (**c**), scatter plots of trends (°C/decade) by elevation for annual mean temperature (**d**) and elevation dependence of annual mean temperature trends (°C/decade) (**e**) in the TP and TPS during1971–2017. (The numbers in horizontal axis in (**e**) represent the stations in each 500 m elevation band, and trend magnitude is plotted on the y axis).

At the same time, the meteorological stations are not evenly distributed in space, especially the lack of meteorological stations in high-altitude areas. It may also be one of the reasons behind the complexity of EDWE signals in extreme precipitation. The extreme precipitation changes with elevation in this region and what are the controlling factors behind the changes need additional study. During the last 50 years, climate change-related warming and frequent extreme climate events have occurred in the TP and TPS [36,61–63]. EDWE has led to significant changes in the cryosphere and increased geohazard risks in mountainous regions [26,60,62–64]. Furthermore, the effects of EDW on EDWE over various time scales remain unclear and require further evaluation.

Research has shown that warming and the frequency of extreme climate events are enhanced in the TP [34], the glaciers that feed glacial lakes are melting and retreating with warming, causing flood outbursts [62,64,65], and these have increased in both frequency and scale [29,48,66]. The EDWE, especially extreme precipitation changes with elevation, may increase the risk of natural disasters. Natural disasters that affect the TP have possible

knock-on effects, which both extend and amplify their damage [64]). In future research, extreme climate events should be analyzed more deeply and their impact on the environment and human society should be evaluated.

## 5. Conclusions

Based on 113 meteorological stations, seven indices of extreme precipitation were used to analyze the spatiotemporal changes in precipitation and extreme precipitation in the TP and TPS from 1971–2017. The relationships between precipitation (extreme precipitation) trends and elevation were examined. The main findings are summarized as follows:

(1)  The annual total precipitation (RR ≥ 1 mm) increased at a rate of 6.7 mm/decade. The spatial distributions of long-term PRCPTOT trends exhibited regional differences. The total amount of extreme precipitation (R95p) and the contribution of extreme precipitation increased, the climate extreme was enhanced.

(2)  The annual and seasonal precipitation trends and the extreme precipitation trend were found to be mostly elevation-dependent, but the trend fluctuated at different elevations. A nonlinear relationship between wetting rates and elevation was observed.

(3)  Each extreme precipitation index showed a different trend change with elevation. The results indicate that extreme precipitation dependence on elevation in mountainous areas is complex and requires further investigation.

**Author Contributions:** Conceptualization, W.H. and J.Y.; writing—original draft preparation, W.H. and J.Y.; Writing—review and editing, Q.H.; Funding acquisition, W.H. and J.Y.; methodology, J.C. All authors have read and agreed to the published version of the manuscript.

**Funding:** This research was funded by the National Natural Science Foundation of China [U1903113, U1903208], the Second Tibetan Plateau Scientific Expedition and Research Program [2019QZKK010206], China Postdoctoral Science Foundation [2019M653905XB], Central Asia Atmospheric Science Research Fund [CAAS201703], and Sichuan Science and Technology Program.

**Data Availability Statement:** Data are available upon reasonable request to the corresponding author.

**Acknowledgments:** We are grateful to the reviewers and for their valuable comments on this manuscript, we also thank the editor for his contribution to the manuscript.

**Conflicts of Interest:** The authors declare no conflict of interest.

## Appendix A. Calculation Method of Extreme Precipitation Indices

1.  RX1day

Let $RR_{ij}$ be the daily precipitation amount on day $i$ in period $j$. Then maximum 1-day values for period $j$ are:

$$\text{Rx1}day_j = \max\left(RR_{ij}\right) \tag{A1}$$

2.  SDII

Let $RR_{wj}$ be the daily precipitation amount on wet days, $W(RR \geq 1.0\text{ mm})$. in period $j$. If $W$ represents number of wet days in $j$, then:

$$SDII_j = \frac{\sum\limits_{w=1}^{w} RR_{wj}}{W} \tag{A2}$$

3.  R10

Let $RR_{ij}$ be the daily precipitation amount on day $i$ in period $j$. Count the number of days where:

$$RR_{ij} \geq 10\text{ mm} \tag{A3}$$

4.  CDD

Let $RR_{ij}$ be the daily precipitation amount on day $i$ in period $j$. Count the largest number of consecutive days where:

$$RR_{ij} < 1 \text{ mm} \tag{A4}$$

5.   CWD *

Let $RR_{ij}$ be the daily precipitation amount on day $i$ in period $j$. Count the largest number of consecutive days where:

$$RR_{ij} \geq 1 \text{ mm} \tag{A5}$$

6.   R95pTOT

Let $RR_{wj}$ be the daily precipitation amount on a wet day $w(RR \geq 1.0\text{mm})$ in period $j$ and let $RR_{wn}95$ be the 95th percentile of precipitation on wet days in the 1971–2017 period. If W represents the number of wet days in the period, then:

$$R95_{pj} = \sum_{w=1}^{w} RR_{wj} \text{ where } RR_{wj} > RR_{wn}95 \tag{A6}$$

7.   PRCPTOT

Let $RRij$ be the daily precipitation amount on day $i$ in period $j$. If $i$ represents the number of days in $j$, then

$$\text{PRCPTOT}_j \sum_{i=1}^{I} RR_{ij} \tag{A7}$$

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
