# Peer review of "Elevation-Dependent Trends in Precipitation Observed over and around the Tibetan Plateau from 1971 to 2017"

_water, doi:10.3390/w13202848_

Round 1

Reviewer 1 Report

Dear Authors,

The presented study concerns the determination of the variability of precipitation, in particular extreme precipitation, as one of the indicators of climate change. The research was carried out at the test site in Tibet Plane.

The subject of the research is interesting, in particular, that the more and more frequent periods of extreme precipitation in mountainous areas lead in many cases to landslides, mudslides, and therefore threats to infrastructure and inhabitants. Unfortunately, while reading the manuscript, I have not found a broader reference to many related phenomena that would justify the need to conduct the research presented in the article. After carefully reading the manuscript, I must say that due to the poorly defined research niche and a poorly written introduction, the article does not present any major novelty.

Already the abstract requires significant improvement, it should be written in the style of academic writing. At the beginning there should be an introduction to the research topic, then a short description of your research aim, key findings and, finally, the possibility to implement your research results. Additionally, the introduction section is written very chaotic, no linearity in the text indicates a lack of logical continuity. Why research on elevation-dependent precipitation is so crucial? This must be clarified and strongly pointed out both in the abstract and in the introduction.

Concerning the methodology adopted, you used R ClimDex software in your research. However, what kind of analyses and calculations did you exactly carry out? The methodology section is very weak and poor written so that it is hard for the reader to capture what steps exactly were performed to carry the research experiment.

The research results and their discussion are written relatively correctly. Although it is not fully known how the research was conducted (the methodology chapter requires many corrections), the results of the research and good quality graphics indicate that the authors put a lot of work into the preparation of the article. I believe that the article as it stands cannot be published in Water due to significant flaws and methodological deficiencies. Nevertheless, I would like to encourage the Authors to read the article carefully again and rewrite it so that it is written in the style of academic writing. Then, the article could be potentially resubmitted for further consideration for publication.

Kind regards,

Reviewer  

Reviewer 2 Report

This manuscript examines the relationship between precipitation (and precipitation extremes) trends and elevation in the Tibetan Plateau based on the data from 113 meteorological stations. The author employed different precipitation indices to discuss the strong dependence and complexity between precipitation and elevation in Tibetan Plateau.

As far as I can tell, the results are original. In my option, this is a good piece of work with some interesting information to help understand the climate change as it relates to the precipitation variabilities, and I recommend the manuscript be published with consideration of these minor questions.

  1. Have the authors considered the relationship between precipitation and the air pressure change when talking about the elevation?
  2. Can precipitation results from the 1971-2017 trends give future climate model projections?

Reviewer 3 Report

Line 101 -102 - some redundancy between the objectives.

For all the figures - must be increased the font size (especially in figures 1, 2, 5, 11).

Attention to font size from all the paper.

The paper discusses simple statistics - precipitation and elevation. Some strong objectives are needed. 

Reviewer 4 Report

It is neccecary to take into account the comments

Round 2

Reviewer 1 Report

Dear Authors,

Thank you for your in-depth responses to my inquires. The manuscript has been substantially improved and now is almost suitable for publication in Water. However, before the publication, I would highly advise making an English native speaker revision of the paper to ensure its highest language correctness. 

Kind regards,

Reviewer

Reviewer 3 Report

The references in the Introduction chapter must be put on the Journal format. Please consult the Instructions for authors!